# Percutaneous Electroosmosis of Berberine-Loaded Ca^2+^ Crosslinked Gelatin/Alginate Mixed Hydrogel

**DOI:** 10.3390/polym14235101

**Published:** 2022-11-24

**Authors:** Yinyin Liu, Si Shen, Yifang Wu, Mengmeng Wang, Yongfeng Cheng, Hongmei Xia, Ruoyang Jia, Chang Liu, Yu Wang, Ying Xia, Xiaoman Cheng, Yan Yue, Zili Xie

**Affiliations:** 1College of Pharmacy, Anhui University of Chinese Medicine, No. 350, Long Zi Hu Road, Hefei 230012, China; 2Clinical College of Anhui Medical University, Hefei 230031, China; 3School of Life Science, University of Science and Technology of China, Hefei 230027, China; 4Anhui Province Key Laboratory of Pharmaceutical Preparation Technology and Application, Hefei 230012, China; 5Anhui Institute for Food and Drug Control, Hefei 230051, China

**Keywords:** gelatin/alginate hydrogel, berberine hydrochloride, electroosmotic transdermal

## Abstract

Flexible conductive hydrogel has been driven by scientific breakthroughs and offers a wide variety of applications, including sensors, electronic skins, biomedicine, energy storage, etc. Based on the mixed-ion crosslinking method, gelatin and sodium alginate (Gel–Alg) composite hydrogels were successfully prepared using Ca^2+^ crosslinking. The migration behavior of berberine hydrochloride (BBH) in the matrix network structure of Gel–Alg hydrogel with a certain pore size under an electric field was studied, and the transdermal effect of berberine hydrochloride under an electric field was also studied. The experimental results show that Gel–Alg has good flexibility and conductivity, and electrical stimulation can enhance the transdermal effect of drugs. Gel–Alg composite hydrogel may be a new material with potential application value in future biomedical directions.

## 1. Introduction

Hydrogel is a kind of polymer material with a three-dimensional network, which can absorb a large amount of water or biological fluid [1]. The presence of hydrophilic groups in the polymer, such as amine (-NH_2_), carboxylic acid (-COOH) and hydroxyl group (-OH), give the hydrogel excellent water-locking ability. In addition to its high water-absorption capacity, hydrogel has good biocompatibility, viscoelasticity, ionic conductivity and tunable release properties. Therefore, it has received wide attention in the biomedical field [2,3].

Different hydrogel polymers can be physically or chemically crosslinked with macromolecules such as polysaccharides, peptides and proteins with good mechanical properties [4,5]. In recent years, smart hydrogels with special environmental responsiveness have been manufactured [6,7]. The physical and chemical properties of the hydrogel will respond abruptly and reversibly to changes in environmental factors such as electric field, temperature, pH, light and magnetic field [8,9,10]. In particular, electroresponsive hydrogels show clear advantages in the field of bioelectronics [11,12,13]. In addition, the high water content of conductive hydrogels allows them to act as the transport for biomolecules and chemical molecules, which can effectively modulate electrically to stimulate cellular activity, making them into candidates for drug delivery, cellular delivery and tissue engineering [14,15,16].

A transdermal drug-delivery system is a kind of preparation in which drugs are absorbed through the skin and reach the body for local or systemic treatment, and can avoid the “first-pass effect” in the liver or gastrointestinal tract. It also has many advantages such as high use rate and convenient use. The excellent properties of hydrogels make them widely used in transdermal drug delivery. Because of the rate-limiting barrier of the stratum corneum, the transdermal performance of most drugs is poor, so it is necessary to find a suitable method to improve skin permeability. Electroosmotic iontophoresis is a process of improving drug molecule penetration across the skin and into the body by applying an appropriate direct current to the skin. Electroosmosis is one of the driving forces of drug molecule transport. Cationic and anionic drugs penetrate the skin at the anode and cathode, respectively. Iontophoresis can greatly improve the skin permeability of drugs and prolong the action time of drugs [17,18]. 

Hydrogels can conduct electricity by adding conductive polymers, conductive ions or conductive particles [19,20]. Based on the origin of the matrix in the hydrogel, they can be divided into two categories—natural hydrogel carriers (chitosan, sodium alginate, hyaluronic acid), and synthetic hydrogel carriers. Compared to synthetic polymers, natural polymer-based conductive hydrogels have advantages of non-toxicity, biodegradability, and superior flexibility and conductivity [21]. Therefore, they are more attractive for practical applications [22]. Most natural hydrogels have disadvantages of low mechanical strength and lack of toughness, while synthetic hydrogels lack biocompatibility [23].Therefore, to promote the practical application of hydrogels and improve their poor mechanical properties, construction strategies such as dual networks and interlocking crosslinked networks have been proposed to investigate hydrogels with regard to both their conductivity and their mechanical properties, using two complementary matrices [24]. 

Alginate is a linear polysaccharide polymer derived from the brown algae kelp, consisting of β-D-mannuronic acid and α-L-gulonuronic acid residues linked by β-1,4 glycosidic bonds [25]. Alginate is non-irritating, ionically crosslinked, easily forms hydrogels, acts directly on macrophages in the wound, promotes wound-healing, and is suitable for various biomedical applications [26,27]. However, the mechanical properties of hydrogels formed by a single sodium alginate are poor and can be improved by physical blending and chemical methods. Gelatin is a natural polymeric material derived from collagen hydrolysis [28]. Hydrogel material formed only by gelatin has disadvantages such as brittleness and poor flexibility. It is usually compounded with other materials to improve its mechanical properties [29]. A study designed an oxidized gelatin–sodium alginate hydrogel that conducted electricity and promoted cell adhesion and proliferation, which contributed to electrical stimulation-assisted tissue engineering [30]. We tried to construct a novel composite material using the temperature-controlled crosslinking property of gelatin, and the crosslinking property of sodium alginate and divalent calcium ions. Gelatin and sodium alginate are tightly bonded to form a spatial network structure with certain pore size, which improves the electrical conductivity and flexibility of the hydrogel. When a charged drug such as berberine hydrochloride (BBH) is wrapped in it, its directional motion can be realized under the energized condition.

BBH is a quaternary ammonium salt alkaloid with good antibacterial, anti-inflammatory and other pharmacological activities [31]. However, its low bioavailability limits its efficacy. Therefore, a lot of research has also been done to improve the bioavailability of BBH [32]. The conductive hydrogel prepared in this study was a drug-carrying hydrogel, around which a mild electric current was applied to improve drug entry into adjacent tissues, which could effectively improve the bioavailability of BBH. In this study, the prepared BBH–Gel–Alg was subjected to an electric field and its transdermal effect was investigated. On the one hand, the in vitro drug release properties of BBH–Gel–Alg were examined, and on the other hand, the effect of the electric field on the transdermal drug release effect of BBH–Gel–Alg was also innovatively explored. This new composite hydrogel can be used not only for slow-release drugs, but also as a new material for biopharmaceuticals.

## 2. Materials and Methods

### 2.1. Materials

Gelatin, calcium chloride anhydrous and sodium hydroxide were purchased from Sinopharm Chemical Reagent Co., Ltd. (Shanghai, China). Sodium alginate was obtained from Tianjin Guangfu Fine Chemical Research Institute (Tianjin, China). Berberine hydrochloride was obtained from Xi’an Reain Biotechnology Co., Ltd. (Xi’an, China). Potassium dihydrogen phosphate was purchased from Xilong Scientific Co., Ltd. (Shantou, China).

### 2.2. Establishment of Standard Curve of Berberine Hydrochloride

Accurately weigh 0.0202 g of berberine hydrochloride in a 100 mL volumetric flask, add PBS buffer solution to dilute to a constant volume, and dissolve completely by ultrasound to obtain 200 μg/mL stock solution. Pipette 0.5 mL, 1.0 mL, 1.5 mL, 2 mL, 2.5 mL and 3.0 mL into a 50 mL volumetric flask, add PBS buffer to dilute to the mark and shake well. Then use the PBS buffer solution as a blank control and measure the absorbance by a UV-1000 spectrophotometer (AOE Instruments, Shanghai, China) at 345 nm.

### 2.3. Preparation of BBH–Gel–Alg Hydrogel

We weighed 1.2 g gelatin and 0.4 g sodium alginate into 1.0 mg/mL berberine hydrochloride solution; the mass concentration of gelatin and sodium alginate was 6% and 2%, respectively. It was stirred in a water bath at 45 °C until completely dissolved, and a uniform BBH–Gel–Alg mixed solution was obtained. The bubbles were removed by standing or ultrasonic vibration, and then an equal volume of 2% calcium chloride solution was added for crosslinking for 12 h. Finally, the surface of the hydrogel was washed with deionized water to obtain a BBH–Gel–Alg composite hydrogel. Gel–Alg hydrogel was prepared in the same way by replacing the berberine hydrochloride solution with PBS solution (Figure 1A).

### 2.4. Performance Evaluation of Hydrogels

The content of gelatin was fixed at 4%, with the tensile capacity and elasticity of Gel–Alg hydrogels containing 1%, 2%, 3%, 4% and 5% sodium alginate were determined, respectively. The stretchability and elasticity of Gel–Alg hydrogels with gelatin content of 2%, 4%, 6%, 8% and 10% were also investigated when the content of sodium alginate was 2%. 

### 2.5. Water-Absorption Properties of Hydrogel

A certain weight (W_0_) of Gel–Alg hydrogel was immersed in distilled water at room temperature for 24 h. Then, the hydrogel was taken out, water was removed from the surface of the hydrogel with filter paper and weighed (W_w_). The water-absorption rate (WR%) was calculated using the following formula:(1)$$\mathrm{WR}\;\left(\%\right)=\frac{W_{W}-W_{0}}{W_{0}}\times 100$$

### 2.6. Swelling Property of Hydrogel

The Gel–Alg hydrogel was fully dried and weighed, and the mass was recorded as W_0_. Then, it was put into deionized water at 25 °C to swell for 48 h, the sample was taken out at time point t, the excess water on the surface of hydrogel was absorbed with filter paper, then it was weighed, with the mass is recorded as W_t_, and the swelling ratio (SR%) calculated according to the formula:(2)$$\mathrm{SR}\;\left(\%\right)=\frac{W_{t}-W_{0}}{W_{0}}\times 100$$

### 2.7. Determination of Drug Loading of Hydrogel

We weighed 1.0 g of BBH–Gel–Alg hydrogel into 100 mL of PBS solution, heated it to 50 °C and stirred the solution at a high speed for 6 h, so that the hydrogel was completely damaged, and the drug was completely dissolved out. The absorbance of the supernatant after centrifugation of the solution was measured by the spectrophotometer, and the drug content was calculated. The drug loading (DL) was calculated by the following formula:(3)$$\mathrm{DL}\;\left(\mathrm{mg}/g\right)=\frac{W_{1}}{W_{2}}$$
where W_1_ (mg) is the weight of BBH in hydrogel, and W_2_ (g) is the weight of hydrogel.

### 2.8. In Vitro Release Study of BBH–Gel–Alg Hydrogel

The dialysis membrane with molecular cutoff of 8000–14,000 was placed on the Franz diffusion cell for experiments. We put 1.0 mL of BBH and BBH–Gel–Alg hydrogel into the supply pool, respectively, and placed a small stirrer in the receiving pool and filled with PBS solution. We put the diffusion cell device into a magnetic stirrer, set the temperature to 37 °C, and the stirring speed to be moderate. Sampling was undertaken of 2.0 mL at 10 min, 20 min, 30 min, 1 h, 2 h, 3 h, 4 h, 5 h, 6 h, 7 h, 8 h, 9 h, 10 h, 11 h, 12 h, 24 h, 36 h, 48 h, 60 h, 72 h, and an equal amount of PBS solution was added to the receiving pool after sampling. PBS and blank hydrogel were used as controls. The absorbance of the sample was measured by a spectrophotometer, and the release amount of the drug was calculated according to the standard curve of berberine hydrochloride.

### 2.9. Drug Movement Behavior Study under an Electric Field

Gel–Alg hydrogel was connected to the agarose gel. BBH was used as the diffusion drug, and a complete BBH release-promoting model under electric field force was constructed. At different time points or different voltages, the movement distance of BBH was measured to analyze the movement of drugs in Gel–Alg hydrogels at different ratios (2:1, 3:1 and 4:1) of gelatin and sodium alginate. Skin tissue was obtained according to the method previously reported in our laboratory [33], and then it was added between agarose gel and Gel–Alg hydrogel, and BBH solution or BBH–Gel–Alg was added into the whole of agarose gel, and the drug migration distance was recorded at 0, 1, 2, 4 and 6 h. After 6 h, the Gel–Alg hydrogel was stirred in sections, centrifuged at 6000 rpm/min for 10 min, and the supernatant was taken to measure the absorbance value and calculate the BBH content.

### 2.10. Study of Drug Release Kinetics of BBH–Gel–Alg Hydrogel

According to the release results of BBH in Gel–Alg hydrogel in vitro, the release model was fitted by Origin software with the linear regression method, and the drug release kinetics mechanism was studied. Four models (zero-order, first-order, Higuchi and Hixson–Crowell) were used for fitting, and the formulas are shown in Table 1. The release model of BBH–Gel–Alg was determined by comparing the correlation coefficients of each equation.

### 2.11. Statistical Analysis

The data results were expressed as mean ± standard deviation (SD). Significant differences were determined using SPSS, version 23.0 (IBM, Armonk, New York, NY, USA), and one-way ANOVA. It was considered significant when *p* < 0.05.

## 3. Results and Discussion

### 3.1. Standard Curve of Berberine Hydrochloride

The regression equation of the standard curve of berberine hydrochloride C = 0.0633*A + 0.0026 (R^2^ = 0.9998). It showed that berberine hydrochloride had a good linear relationship in the range of 2.0~12.0 μg/mL. The RSD of precision and stability were 0.29% and 0.92%, respectively, which indicated that the ultraviolet spectrophotometry method had good precision and stability, and it was an effective method to detect the content of berberine hydrochloride.

### 3.2. Properties of Composite Hydrogels

When the gelatin concentration was constant, the viscosity of the mixed solution increased with the increase of the sodium alginate concentration [34]. Since the viscosity depended on the concentration of each component of the solution and the degree of intermolecular bonding, and since both gelatin and sodium alginate are hydrophilic substances, and the two are miscible, more bonding occurs with each other, and interactions such as hydrogen bonding and electrostatic interactions also occur, increasing the viscosity of the mixed solution [35]. When the calcium chloride concentration of the crosslinking agent was 2%, as the sodium alginate concentration increased, the color of the composite gel was milky white, the transparency decreased, and the formability was poor [36,37]. The morphologies of the hydrogels with different contents of sodium alginate are shown in Figure 2A. Sodium alginate and calcium ions could crosslink quickly. When the concentration of sodium alginate was high, the crosslinking between gelatin molecules and sodium alginate molecules became close, and the viscosity of the colloid was large, which was not conducive to the crosslinking of calcium ions, thus affecting the formability of the composite hydrogel. With the increase of gelatin concentration, the transparency of the composite hydrogel increased, and the colloid was dense, high in hardness, and had good mechanical properties. The increase of gelatin concentration could effectively reduce the loss of water, so that the hydrogel had good water-holding performance. The morphologies of hydrogels containing different concentrations of gelatin are shown in Figure 2B. However, when the gelatin concentration was too high, the elasticity and toughness were relatively poor, and the stretching was easy to break. 

Gelatin and sodium alginate interacted via the electrostatic force between the side chain amino group of gelatin and the carboxyl group of sodium alginate, as well as the intermolecular force, hydrogen bond, coordination bond and others [38]. Adding natural polymer sodium alginate to gelatin could make use of the good hydrogel property of gelatin itself and the free amino groups on its components to form an interpenetrating network hydrogel with sodium alginate, increasing the hydrogen bonding force in the hydrogel [39]. The formation of a more uniform three-dimensional network structure is shown in Figure 1B–D. When calcium ions were added to the network structure formed by the crosslinking of sodium alginate and gelatin, Ca^2+^ was easy to combine with the α-L-guluronic acid (G-region) of sodium alginate, connecting the G-region of long chain molecules to form a stable spatial structure, and the molecules were crosslinked using a coordination bond to form a stable hydrogel system [40]. Ca^2+^ competes with gelatin for the carboxyl group on the surface of sodium alginate, making the pore size of hydrogel more compact, and the elasticity and toughness of hydrogel increases as the brittleness decreases. The crosslinking of Ca^2+^ made the combination of sodium alginate and gelatin firmer and tighter, and the hydrogel had a certain pore size and a more stable spatial network structure [41]. Finally, the viscosity, strength, elasticity and pore size of the composite hydrogel prepared with the concentration of 2% sodium alginate and 6% gelatin all met the requirements of flexible conductive hydrogel. The flexibility, adhesion and hardness of the hydrogel are shown in Figure 2C.

The fluid that satisfies the linear relationship between shear stress and shear strain rate is called Newtonian fluid, and the fluid that does not satisfy the linear relationship is called non-Newtonian fluid [42,43]. The composite hydrogel studied in this paper belongs to a non-Newtonian fluid. From the perspective of non-Newtonian fluid, when the proportion of sodium alginate increases, the viscosity of its colloid increases. The main reason for the increase in viscosity is that as the proportion of sodium alginate increases, Ca^2+^ in solution gradually collects in the hydrogel, the number of positive ions in the molecular chains increases, and the electrostatic repulsive force weakens. The stretched state of the molecular chains gradually changes to a curled and aggregated state, gradually forming a “local network” structure and finally a complex network structure. When the proportion of sodium alginate increases, the crosslinking between the molecules of gelatin and sodium alginate becomes closer, so its viscosity increases. However, when the proportion of gelatin is higher than 10%, its elasticity and toughness deteriorate, mainly because the dense pore size increases the water retention capacity of hydrogel, so its elasticity and toughness diminish. 

### 3.3. Water-Absorption Properties of Gel–Alg Hydrogel

The effect of different contents of sodium alginate and gelatin on the water absorption of Gel–Alg hydrogel in distilled water is shown in Figure 3. With the increase of sodium alginate content, the water-absorption rate of Gel–Alg hydrogel increased continuously. When the sodium alginate content was 5%, the water-absorption rate was about 100%, reaching the maximum value. The reason for this might be that the hydrogel prepared using sodium alginate with proper mass ratio had the best-structured three-dimensional network formed by a crosslinking agent, and could hold more water molecules. With the increase of gelatin content, the water-absorption rate of Gel–Alg hydrogel showed a slow downward trend, and when the gelatin content was 6%, the water-absorption rate was about 20%. The hydrogel with sodium alginate concentration of 2% and gelatin concentration of 6% had the lowest water-absorption rate, which could keep a good shape after absorbing water, and it was beneficial to the drug release behavior.

### 3.4. Swelling Properties of Gel–Alg Hydrogel

The swelling capacity of hydrogel is one of the important factors when studying its application performance in tissue engineering and drug-delivery systems. The swelling ratio of Gel–Alg hydrogels with different contents of sodium alginate and gelatin changed with time, as shown in Figure 4. The swelling ratio of all hydrogels increased continuously across 12 h, and then remained constant. These results indicated that Gel–Alg hydrogel reached a swelling equilibrium after 12 h. A certain amount of water was absorbed and retained in these hydrogel structures. The swelling ratio of Gel–Alg hydrogel increased with the increase of sodium alginate content, but the swelling rate reached the highest when the gelatin content was 2%. The relationship between the content of gelatin and swelling ratio was complicated, and there was no content-dependent relationship.

### 3.5. The Study on Migration of Berberine Hydrochloride under an Electric Field

As shown in Figure 5A, the migration distance of berberine in Gel–Alg hydrogel was affected by voltage and gelatin content. With the electric field strength improving, the force received by the positively charged berberine in the electric field increased, therefore increasing the migration distance of berberine [44]. When the voltage was 180 V and 200 V, the migration distance of BBH in the Gel–Alg (4:1) was the shortest, which indicated that the higher the concentration of gelatin in Gel–Alg, the smaller the internal aperture of hydrogel, which made the berberine hydrochloride move slowly in the Gel–Alg hydrogel. Additionally, when the voltage was 220 V, the migration distance of BBH in the Gel–Alg (2:1) was the longest, which might be due to the larger pores of the Gel–Alg hydrogel. The smaller the resistance to berberine hydrochloride, the greater the displacement. 

The time and migration distance in the Gel–Alg (4:1) showed a linear relationship as shown in Figure 5B. To determine the release mechanism of berberine hydrochloride (positively charged drug) in Gel–Alg (4:1), four models were studied, namely zero-order, first-order, Higuchi and Hixson–Crowell models [45]. The correlation coefficient R^2^ of each model simulation curve is shown in Table 2. The R^2^ of zero-order model of drug release is relatively high, indicating that the drug release mechanism of berberine hydrochloride seems to be more consistent with the zero-order kinetics study.

### 3.6. The Release of Berberine Hydrochloride In Vitro

The in vitro release behavior of berberine hydrochloride was investigated based on the BBH–Gel–Alg hydrogel prepared in this experiment with a loading capacity of 0.87 mg/g. The graph with time as abscissa and cumulative drug release rate as ordinate is shown in Figure 6. The release of berberine hydrochloride from Gel–Alg hydrogel in PBS buffer solution at 37 °C (pH = 7.4) was studied in this experiment. After 72 h of simulated release in vitro, it was found that the cumulative release rate of BBH reached 60 ± 1.24% within 7 h, and then became stable and the release rate was faster. However, BBH-loaded Gel–Alg hydrogel released slowly within 72 h, and the final cumulative release rate was about 50 ± 1.68%. The drug release kinetics of BBH and BBH–Gel–Alg were studied by zero-order, first-order and Higuchi models, respectively. The correlation coefficient (R^2^) was the best parameter to choose the model [46]. Table 3 showed the fitting results of cumulative drug release rate of BBH and BBH–Gel–Alg. The results showed that the release of BBH and BBH–Gel–Alg were in accordance with the first-order release model, i.e., the drug was released according to a constant proportion per unit of time.

Compared with BBH, BBH–Gel–Alg had good sustained release effect. This was because in the process of drug release, berberine was wrapped in the network structure of hydrogel, and its release path was more tortuous. Because hydrogels contain many hydrophilic groups, they could be used as crosslinking sites for non-covalent bonding with berberine, so that berberine could be connected into the network structure of the hydrogel [47]. The slow release of BBH–Gel–Alg also proved that the hydrogel formed a complex network structure, which was beneficial to drug loading. Adding natural polymer sodium alginate to gelatin could make use of the good hydrogel property of gelatin itself and the free amino groups on its components to form an interpenetrating network hydrogel with sodium alginate increased the hydrogen bonding force in the hydrogel, and formed a more uniform three-dimensional network structure [48]. In addition, we found that BBH–Gel–Alg was pH responsive by consulting a large amount of literature. Under weak acid conditions, the drug release rate tends to accelerate [49]. Considering that the hydrogen bond, Van Der Waals force, inter component force, electrostatic influence, etc. connected between the composite hydrogels were affected and degraded, resulting in the accelerated drug release rate, the stronger the acidity, the faster the drug release [50]. Generally speaking, Gel–Alg hydrogel as the sustained release preparation has the advantages of keeping a stable blood drug concentration, good efficacy, convenient medication, and few side effects [51].

### 3.7. The Results of Berberine Hydrochloride Transdermal under an Electric Field

We established a model that stimulated the transdermal release of berberine hydrochloride from Gel–Alg hydrogel under the action of an electric field (Figure 7A). As shown in Figure 7B, we recorded the transdermal migration distance of berberine hydrochloride within 6 h and plotted the curve between migration distance and time. According to Figure 7C, we found that the direct migration of berberine hydrochloride was close to transdermal migration within 2 h. After 2 h, there was a difference between direct release and transdermal release. In addition, after 4 h of power, we observed the uneven distribution on the Gel–Alg hydrogel surface. The clear yellow color on the surface of the hydrogel near the positive end was due to the fluorescence property of BBH itself, while the white turbid precipitate appeared on the hydrogel near the negative end, which might be because calcium ions also moved with the current, but the appearance of white turbidity was attributed to calcium ions being insoluble in PBS. This also showed that the existence of Ca^2+^ made the double network structure of hydrogel more compact [52].

After the end of electrical stimulation, we measured the content of BBH that penetrated the skin. It was found that under electrical stimulation, the proportion of BBH that penetrated the skin within 6 h was 26.25 ± 1.97%. However, in 72 h, the drug amount of BBH transdermal release was about 20 ± 1.54%. Although there was no significant difference in drug release, the electric field could accelerate the permeation rate of berberine hydrochloride. This might also be because BBH is a highly alkaline quaternary ammonium salt with positive charge. Under the action of electrostatic attraction, BBH would move directionally towards the negative electrode and penetrate the skin in a short time [53,54]. In addition, in the absence of an electric field, the diffusion of berberine mainly depended on the diffusion of solution concentration difference [55,56]. The addition of electric current could stimulate the permeability across the skin cells and facilitate the entry of charged compounds (Figure 7D) due to temporary membrane pores being formed under electrical stimulation. In previous reports, the bioavailability of berberine hydrochloride is low, and the amount of BBH across the skin increased with the electroosmosis, iontophoresis and other means, aimed at treating some skin diseases, such as specific dermatitis, psoriasis and so on.

## 4. Conclusions

Gel–Alg composite hydrogels with a double network structure were prepared using gelatin and sodium alginate as the matrix and Ca^2+^ as the crosslinking agent, and their basic properties were investigated. The experimental results showed that Gel–Alg has a uniform pore size, good flexibility, certain elasticity and moderate viscosity. Then, we focused on the in vitro release of BBH–Gel–Alg and the transdermal effect of electrical stimulation. In vitro release experiments showed that a BBH solution and BBH–Gel–Alg conformed to the primary release equation, and had a certain slow-release performance as drug release carriers. Under electrical stimulation, both BBH solution and BBH–Gel–Alg significantly increased their transdermal velocity. Therefore, we inferred that electrical stimulation was the key factor to improving the skin penetration of berberine. In conclusion, Gel–Alg composite hydrogel can not only be used as a carrier for slow-release drugs, but its efficient delivery of berberine hydrochloride under electrical stimulation also makes it a highly efficient transdermal material, and shows potential value.

## Figures and Tables

**Figure 1 polymers-14-05101-f001:**
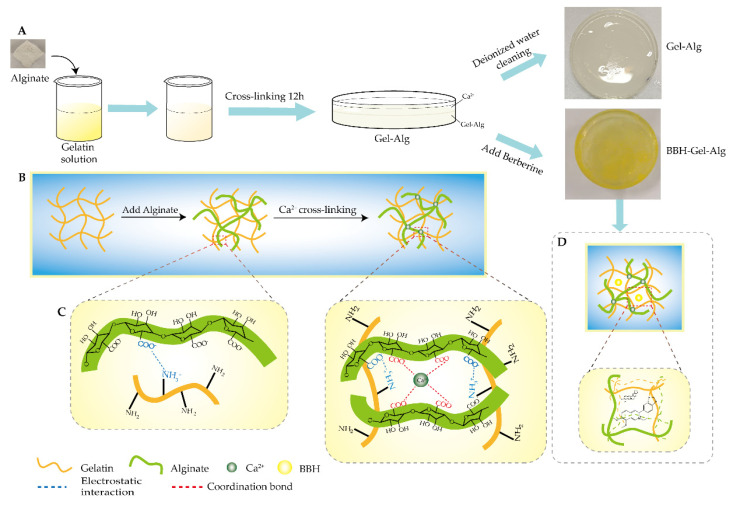
Preparation diagram of Gel–Alg hydrogel and double-layer network structure and interaction force of Gel–Alg hydrogel. (**A**) Preparation process of Gel–Alg hydrogel and BBH–Gel–Alg hydrogel; (**B**) Complex double-layer network structure formed by crosslinking of gelatin, sodium alginate and calcium ions; (**C**) Electrostatic interaction between gelatin and sodium alginate, and the coordination bond between Ca^2+^ and sodium alginate; (**D**) Double-layer network structure and three-dimensional structure of BBH–Gel–Alg hydrogel.

**Figure 2 polymers-14-05101-f002:**
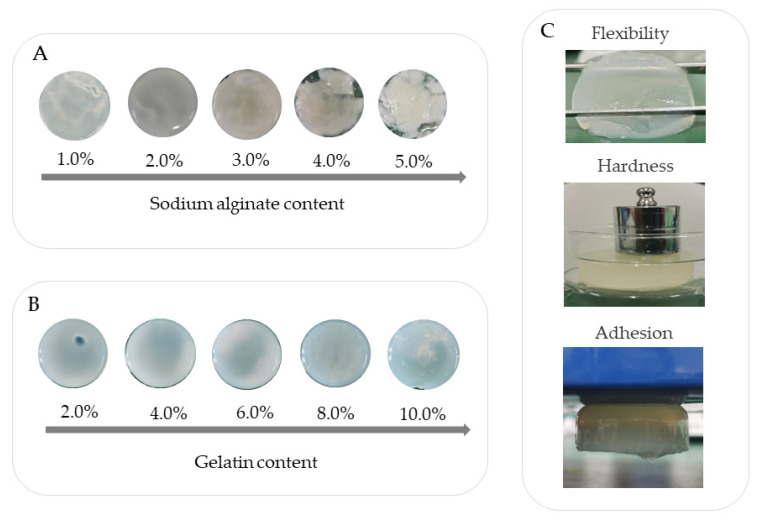
Properties of Gel–Alg hydrogel. (**A**) Morphology of Gel–Alg hydrogel containing different concentrations of sodium alginate. (**B**) Morphology of Gel–Alg hydrogel containing different concentrations of gelatin. (**C**) Flexibility, stiffness and adhesion performance of Gel–Alg hydrogel.

**Figure 3 polymers-14-05101-f003:**
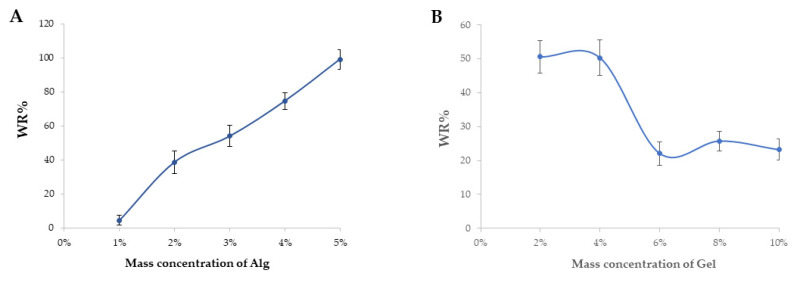
Water absorption of (**A**) sodium alginate and (**B**) gelatin with different mass concentrations.

**Figure 4 polymers-14-05101-f004:**
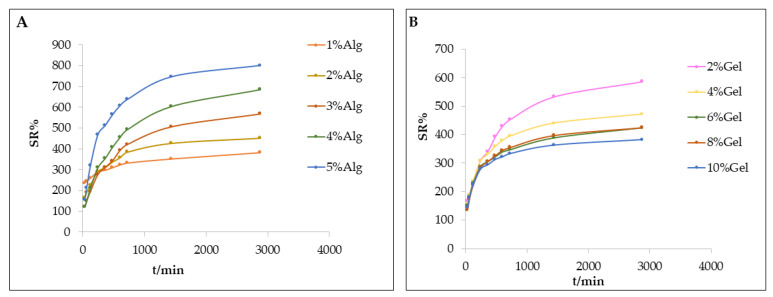
Effect of different ratios of (**A**) sodium alginate and (**B**) gelatin on the swelling property of composite hydrogel.

**Figure 5 polymers-14-05101-f005:**
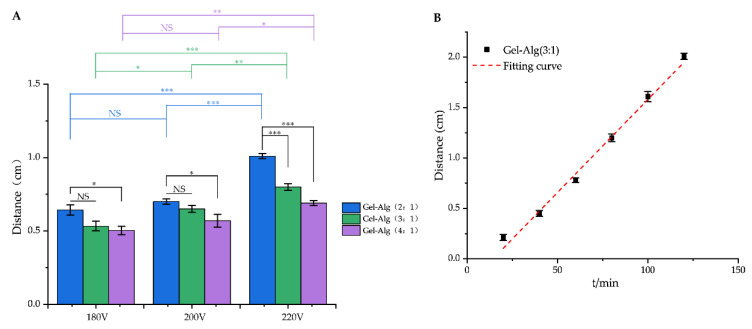
The migration behavior of berberine under electric field. (**A**) The migration distance of berberine hydrochloride in the Gel–Alg hydrogel with different proportions (2:1, 3:1 and 4:1) under different voltages. The data are means ± SD (*n* = 3), and statistical comparisons were made using one-way ANOVA followed by Duncan’s test (NS *p* > 0.05, * *p* < 0.05, ** *p* < 0.01, *** *p* < 0.001. Black represents the statistical difference between each condition verse Gel–Alg (2:1) at the same voltage; blue represents the statistical difference of Gel–Alg (2:1) between different voltages; green represents the statistical difference of Gel–Alg (3:1) between different voltages; purple represents the statistical difference of Gel–Alg (4:1) between different voltages); (**B**) Migration distance and fitting curve berberine hydrochloride in the Gel–Alg (3:1) hydrogel at different time points.

**Figure 6 polymers-14-05101-f006:**
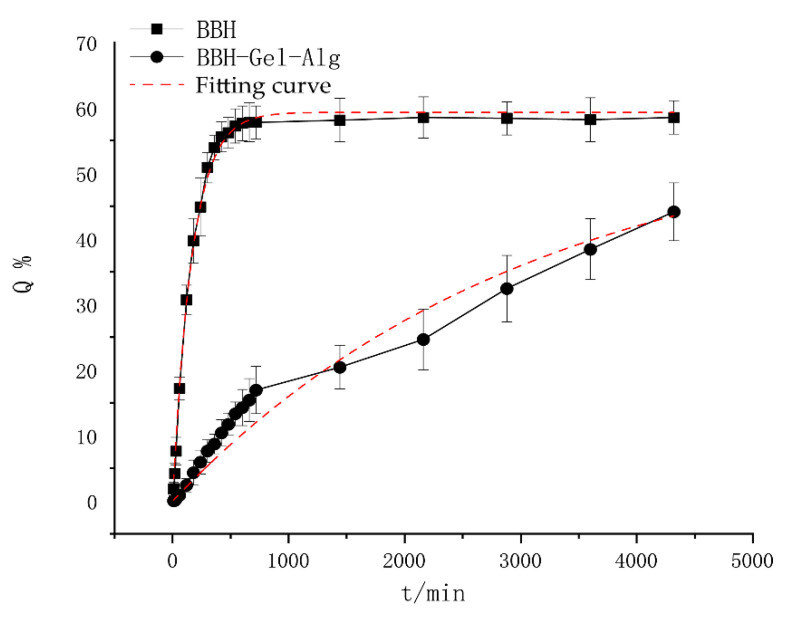
The release of BBH and BBH–Gel–Alg at different times in vitro.

**Figure 7 polymers-14-05101-f007:**
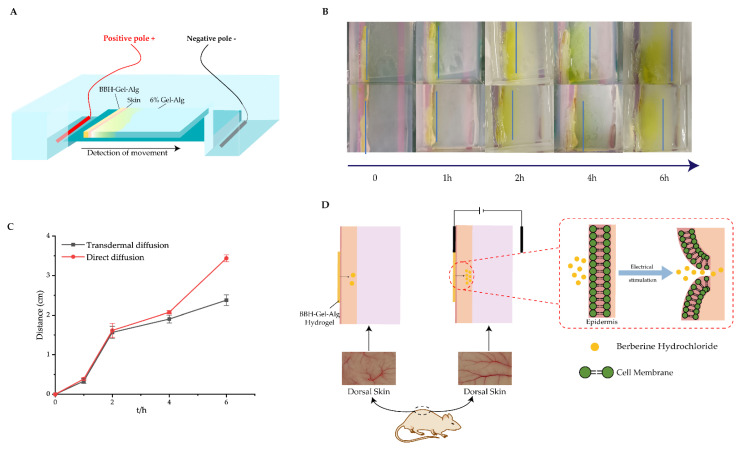
The results and schematic diagram of berberine hydrochloride migration across the skin tissue under electrical stimulation. (**A**) Schematic diagram of hydrogel electrophoresis apparatus; (**B**) The migration behavior (the distance) of BBH–Gel–Alg with times; (**C**) the moving distance of BBH–Gel–Alg with time under power on and power off conditions (Line chart); (**D**) Transdermal electroosmosis mechanism of BBH-loaded Gel–Alg hydrogel.

**Table 1 polymers-14-05101-t001:** The formula of drug release kinetics model.

Model	Formula
Zero-order	Q = a + bt
First-order	Q = a*(1 − e^−bt^)
Higuchi	Q = at^1/2^ + b
Hixson–Crowell	Q = 100[1 − (1 − at)^3^]

“Q” is the cumulative release rate, “t” is the time, and “a” and “b” are the drug release rate constants.

**Table 2 polymers-14-05101-t002:** The simulation results of dynamic model of Gel–Alg (3:1).

Formula	Zero-Order	First-Order	Higuchi	Hixson–Crowell
Gel–Alg (3:1)	D = 0.0177t − 0.1233	D = 12.207 (1-e^−0.0015t^)	D = 0.239t^1/2^ − 0.746	D = 100[1-(1 − 5.58 × 10^−5^t)^3^]
	R^2^ =0.9924	R^2^ =0.9634	R^2^ =0.9633	R^2^ =0.9850

**Table 3 polymers-14-05101-t003:** Release kinetic formulas of BBH and BBH–Gel–Alg.

Formula	Zero-Order	First-Order	Higuchi
BBH	Q = 0.0081t + 36.54	Q = 59.32 (1-e^−0.0059t^)	Q = 0.763t^1/2^ + 25.02
	R^2^ = 0.2549	R^2^ = 0.9968	R^2^ = 0.4832
BBH–Gel–Alg	Q = 0.0097t + 4.262	Q = 45.45 (1-e^−0.0004t^)	Q = 0.692t^1/2^ − 3.77
	R^2^ = 0.9384	R^2^ = 0.9904	R^2^ = 0.9862

## Data Availability

Not applicable.

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
