# Peer review of "Percutaneous Electroosmosis of Berberine-Loaded Ca2+ Crosslinked Gelatin/Alginate Mixed Hydrogel"

_polymers, 2022, doi:10.3390/polym14235101_

Round 1

Reviewer 1 Report

Authors fabricated gelatin and alginate hydrogel and crosslinking with calcium ion under electric field. However, this manuscript lacks novelty and written very poorly. Results and discussion are not sufficient to support the material part. Drug migration behaviour was not studied properly. Additional experiment need. Therefore, current form of manuscript is not suitable for the publication in Polymers journal.

In title “Bebavior” spelling error? Please correct it.

Make spaces between word and citations. For ex: line 40 and other places in the manuscript. Check carefully.

Line 56, need citations.

Line 102, 104, certain masses what does it mean? Please indicate exact weight for readers to reproduce the data?

How authors know the interaction was done between drug, gelatine and alginate and Ca2+? Please supply related data? At least FTIR data and NMR data?

Swelling expt. is missing.

Cite the following references in the manuscript accordingly; Polymer Engineering and Science 62(5) (2022) 1526; https://doi.org/10.1016/j.jmbbm.2017.12.018; Food packaging and Shelf Life, 33 (2022) 100904; https://doi.org/10.1016/j.electacta.2018.05.124.

What about drug encapsulation?

Did authors calculate the crosslinking degree? How authors know crosslinking? Please follow this paper https://doi.org/10.1016/j.msec.2019.02.091.

There is no equation for kinetic study?

Authors written mechanical properties was enhanced after crosslinking where is quantitively mechanical data.

Fig. 3, need statistical analysis for comparison

Line 331, authors mention “improve its bioavailability” where is bioavailability data in the manuscript?

Line 323, where is basic properties. I do not any basic properties authors studied in the manuscript for the crosslinking hydrogel or drug loading crosslink hydrogel.

References style is different make it consistency and follow journal guidelines. For ex. Authors use journal name in abbreviation in ref. no 38 but in 39 full names.

What are the limitations of this study. No indication in the manuscript.

Please comapre this study with other recent similar studies. perferable in the tabular format.

English also poor.

Reviewer 2 Report

Comments:
The manuscript “Migration Bebavior of Berberine in Gelatin/Alginate Blended  Hydrogel Cross-linked by Ca2+ under the Electric Field” by Zili Xie and colleagues introduced gelatin and sodium alginate (Gel-Alg) composite hydrogel prepared by Ca2+ crosslinking. The elasticity, viscosity, flexibility was evaluated. Berberine hydrochloride (BBH) was successfully incorporated into the hydrogel network. And the migration behavior of BBH and BBH-Gel-Alg was assessed.  However, the reviewer believes that additional points of clarifications could potentially be addressed to further strengthen the manuscript.

1.     In line 22, the authors need to put the abbreviation BBH behind berberine hydrochloride, then you can use this abbreviation in line 24.

2.     In line 95, ‘100mL’ need to correct to ‘100 mL’.

3.     In line 97, ‘0.5mL, 1.0mL, 1.5mL, 2mL. 2.5mL’, need to correct to ‘0.5 mL, 1.0 mL, 1.5 mL, 2 mL, 2.5 mL’.

4.     In Figure 1, the scheme showed the formation of the BBH-Gel-Alg composite hydrogel, but there was no BBH information included in this scheme. I would recommend add BBH in this scheme to make it clearer.

5.     In line 126, to keep it consistent, ‘Sampling 2ml’ needs to be corrected to ‘2 mL’.

6.     In line 176, the author said, ‘the gelatin concentration was too high, the elasticity and toughness were relatively poor’. Does the author design any experiment to prove this? Also the concentration was too high seems ambiguous, the author should provide the exact number of what the gelatin concentration can cause the poor elasticity and toughness.

7.     From line 188 to 190, the author states that the Ca2+ make the pore size of the hydrogel more compact, I think the author should provide some SEM data to show the pore size become more compact by adding Ca2+.

8.     In Figure 3A, the author compared the migration distance of BBH in the Gel-Alg at different ratio (1:4 to 1:2) under 180 V, 200 V and 200 V. I would recommend the author add statistical analysis in Figure 3A to make this clearer.

9.     In line 310, to keep it consistent, Figure 5C needs to be bold to Figure 5C.

10.  The author also needs to mention Figure 5A, B, D in the manuscript, if the author didn’t discuss those data, they shouldn’t be included in the manuscript.

11.  The author stated the Gel-Alg hydrogel synthesized by adding Ca2+, but alginate can form hydrogel itself with Ca2+. Can you explain the difference between alginate hydrogel and the Gel-Alg hydrogel?

Reviewer 3 Report

The article addresses a technology that certainly has a significant impact in the improvement of bio-hybrid devices for biomedical applications. Presented results seem to support the future use of the Gel-Alg hydrogel in the above mentioned area of application. The manuscript may be considered for publication in Polymers provided that the authors satisfactorily respond to the queries listed below.

Major Issues

The experimental setup shown in Figure 7 is characterized by a high level of complexity because of the multi-physical phenomena that are involved in the motion of the BBH throughout the hydrogel. It would be interesting to consider the use of mathematical models and simulation tools in order to investigate, for example:

1. the effect of the electric field on BBH velocity as a function of the applied voltage drop;

2. the role of diffusivity on BBH motion.

The authors are invited to consider the issue of using simulation techniques based on mathematical models in order to accompany the experimental analysis with a theoretical approach that allows to optimize the design of the hydrogel.

Minor Issues

Page 4, lines 134-135.

Calculate the water absorption rate (WR%) according to the following formular: ----------> The water absorption rate (WR%)was calculated using the following formula:

lines 142-143.

is calculated according to the formular: ---------->

is calculated according to the formula:

line 151: the following formular: -------> the following formula:

Round 2

Reviewer 1 Report

The authors have not revised any comments properly. No Improvement in this submission. 

I strongly suggest to the authors withdraw the manuscript and revise carefully all the comments 100% and resubmit it to this journal later.

Reviewer 2 Report

The author has addressed my comments carefully and well explained the BBH-Gel-Alg. I think now this manuscript is well-written and can be accepted by MDPI polymers. 

Reviewer 3 Report

The authors have satisfactorily addressed the questions and corrections contained in the previous review. As such, the revised manuscript can be considered for publication in Polymers.

Round 3

Reviewer 1 Report

In the second round of review authors still have not improved the manuscript. I have the same question how do authors confirm the preparation of the material without identifying the chemical interaction and related techniques such as FTIR and NMR?

The authors have not performed any test in vivo on how authors added the mouse image in fig. 7. 

How do authors know the crosslinking was done no expt. evidence.

authors mentioned “improve its bioavailability” where is bioavailability data in the manuscript?

For other minor comments, please check my comments in round 1.